# Classifying Pituitary Adenoma Invasiveness Based on Radiological, Surgical and Histological Features: A Retrospective Assessment of 903 Cases

**DOI:** 10.3390/jcm11092464

**Published:** 2022-04-27

**Authors:** Liang Lu, Xueyan Wan, Yu Xu, Juan Chen, Kai Shu, Ting Lei

**Affiliations:** 1Department of Neurosurgery, Tongji Hospital, Tongji Medical College, Huazhong University of Science and Technology, Wuhan 430030, China; tjluliang@tjh.tjmu.edu.cn (L.L.); xywan@tjh.tjmu.edu.cn (X.W.); deliaxu@gmail.com (Y.X.); jchen@tjh.tjmu.edu.cn (J.C.); kshu@tjh.tjmu.edu.cn (K.S.); 2Sino-German Neuro-Oncology Molecular Laboratory, Tongji Hospital, Tongji Medical College, Huazhong University of Science and Technology, Wuhan 430030, China

**Keywords:** pituitary adenoma, invasiveness, classification, cavernous sinus, radiology, surgery, histology

## Abstract

Invasiveness is a major predictor of surgical outcome and long-term prognosis in patients with pituitary adenomas (PAs). We assessed PA invasiveness via radiological, surgical and histological perspectives to establish a classification scheme for predicting invasive behavior and poor prognosis. We retrospectively analyzed 903 patients who underwent transnasal-transsphenoidal surgery between January 2013 and December 2019. Radiological (hazard ratio (HR) 5.11, 95% confidence interval (CI): 3.98–6.57, *p* < 0.001) and surgical (HR 6.40, 95% CI: 5.09–8.06, *p* < 0.001) invasiveness better predicted gross-total resection (GTR) and recurrence/progression-free survival (RPFS) rates than did histological invasiveness (HR 1.44, 95% CI: 1.14–1.81, *p* = 0.003). Knosp grades 2 (HR 4.63, 95% CI: 2.13–10.06, *p* < 0.001) and 3 (HR 2.23, 95% CI: 1.39–3.59, *p* = 0.011) with surgical invasiveness were better predictors of prognosis than corresponding Knosp grades without surgical invasiveness. Classifications 1 and 2 were established based on radiological, surgical and histological invasiveness, and Knosp classification and surgical invasiveness, respectively. Classification 2 predicted RPFS better than Knosp classification and Classification 1. Overall, radiological and surgical invasiveness were clinically valuable as prognostic predictors. The convenience and good accuracy of Invasiveness in Classification 2 is useful for identifying invasive PAs and facilitating the development of treatment plans.

## 1. Introduction

Pituitary adenomas (PAs) comprise 10–20% of all brain tumors, with a prevalence rate of 80–100 per 100,000 person-year [1,2]. Although PAs are benign adenomas, invasive PA behavior, especially within the cavernous sinus encasing the internal carotid artery (ICA), may lead to surgical failure, progression, and recurrence [3,4,5,6,7,8]. Complications caused by repeated treatment for invasive PAs significantly affect patient quality of life and mortality [9]. Therefore, improved methods for identifying invasive PAs are needed.

Typically, invasive PAs are identified via preoperative radiology, such as magnetic resonance imaging (MRI), using Knosp classification to predict surgical invasiveness; however, the Knosp classification alone is widely used as an accurate predictor of prognosis nowadays [3,10]. The use of endoscopy has expanded, thereby facilitating the direct observation of PAs during an operation and improving the accuracy of recognizing surgical invasiveness [11,12]. It has been suggested that endoscopy may be used to distinguish invasion from compression, a task that is difficult via radiological evaluation [3,12]. Both MRI and endoscopy techniques assess PA invasiveness at the macro-level. In contrast, histological assessment of basal sellar dura samples has been used to assess PA invasiveness at the micro-level, thereby improving our understanding of the histological characteristics of invasive PAs [13,14].

A drawback of the Knosp classification method is its tendency to produce false positives; therefore, auxiliary information is needed to accurately predict the presence of invasive PAs [12,13,14,15]. Serioli et al. suggested that the diagnosis of invasive PAs should be based on the comprehensive evaluation of radiological, surgical, and histological findings [16]. To the best of our knowledge, a comparative analysis of these types of predictors of invasiveness has not been performed. This study aims to compare three PA evaluation strategies to facilitate the establishment of a classification system of PA invasiveness, based on radiological, surgical, and histological perspectives. The system will help distinguish PAs with invasive behavior and a poor prognosis, thereby facilitating individualized treatment planning and appropriate follow-up of patients with invasive PAs.

## 2. Materials and Methods

### 2.1. Study Population and Design

We retrospectively assessed data from patients with PAs who underwent transnasal-transsphenoidal surgery (TTS) at Tongji Hospital between January 2013 and December 2019. Inclusion criteria were as follows: (1) patients with histologically confirmed PAs; (2) patients who underwent microscopic, endoscope-assisted TTS; (3) those who underwent surgery performed by the experienced pituitary surgeon, T.L., and (4) those who were followed up post-surgery for more than 2 years. The following patients were excluded from the analysis: (1) those who did not undergo follow-up evaluations; (2) those who underwent a second surgery during hospitalization; (3) those with no data related to PA invasiveness, and (4) those treated with chemotherapy or radiotherapy. In total, 903 patients (396 female and 507 male) were included in the study. A flowchart summarizing the enrollment strategy and study design is shown in Figure 1.

Gross-total resection (GTR) was defined as the absence of a visible tumor 3–6 months post-surgery via MRI. PA recurrence was defined as the radiological reappearance of a PA after GTR or an increase in plasma hormone levels post-remission. PA progression was defined as tumor regrowth visible via MRI or an increase in plasma hormone levels. Recurrence/progression-free survival (RPFS) in patients with PAs was defined as the duration between TTS and PA recurrence/progression. The recurrence/progression status of each patient was evaluated via follow-up for at least 2 years following surgery. MRI images and endocrine data were obtained via regular outpatient visits (3 months post-surgery and annually thereafter). This study was approved by the Ethics Committee of Tongji Hospital (TJ-IRB20211166). Due to the retrospective nature of the cohort study, the need for informed consent was waived.

### 2.2. Radiological Evaluation

For each patient, pre-and postoperative T1- and T2-weighted MRI images, with and without contrast enhancement, were collected using standard 1.5- or 3.0-T scanners. Radiological indicators of PA invasiveness were evaluated via Knosp grading of coronal T1-weighted contrast imaging data [3,10,17]: Knosp 0, PAs do not pass the tangent of the medial aspects of the intracavernous and supracavernous ICAs; Knosp 1, PAs pass the medial tangent, but do not go beyond the intercarotid line; Knosp 2, PAs extend beyond the intercarotid line, but do not pass the lateral tangent; Knosp 3A, PAs extend lateral into the superior cavernous sinus compartment; Knosp 3B, PAs extend into the inferior cavernous sinus compartment, and Knosp 4, PAs with a total encasement of intracavernous ICA. Knosp grades 3–4 were classified as invasive PA in this study.

### 2.3. Surgical Evaluation

TTS was performed by an experienced pituitary surgeon (T.L.) who has conducted more than 200 TSS procedures annually for the past ten years. Microscopic, endoscopically assisted TTS was performed in all patients considered. The sellar PA was typically removed via a microsurgical approach. The relationship between the PA and the medial wall of cavernous sinus (MWCS) was observed after intra-tumoral decompression. Careful evaluation of whether the PA had infiltrated the cavernous sinus and the relationship between the PA and ICA were performed. Endoscopy was used to expose and excise PAs that had invaded the cavernous sinus and abutted the ICA. In agreement with the opinion of Nishioka [18], we suggest that microscopic techniques are superior to endoscopic techniques for performing fine surgical procedures and effectively controlling venous bleeding; however, endoscopy allows for improved surgical visibility, facilitating the direct visualization of the MWCS. During the close observation, the endoscope was used to reevaluate the integrity of MWCS. The invasiveness of the PA was judged by the pituitary surgeon (T.L.), and his impressions were recorded in detail in the operation report.

Cavernous sinus invasion (CSI) was defined as PA invasion of the MWCS or cavernous sinus, which is observable directly during surgery. Surgical invasiveness refers to invasive and destructive PA growth to the cavernous sinus, sphenoid sinus, ramp and diaphragm sella under direct vision during surgery.

### 2.4. Histological Evaluation

Basal sellar dura samples (endosteum) from the anterior wall were collected for histological examination after sellar floor resection through the sphenoid sinus [14]. All primary TTSs involve the same dural sampling procedure. However, in some patients, we were unable to obtain endosteum due to the small size, invasiveness, or destruction of PAs. Due to safety concerns, MWCS specimens are rarely collected.

Fresh dural samples and PA specimens were fixed with 10% formalin zinc, routinely treated, and embedded in paraffin for immunohistochemical staining. Based on immunohistochemical and clinical findings, PAs were classified as follows: nonfunctioning, prolactin (PRL) secreting, growth hormone (GH) secreting, adrenocorticotropic hormone (ACTH) secreting, thyroid stimulating hormone (TSH) secreting, or plurihormonal PAs. Hematoxylin-eosin-stained dura mater sections were analyzed via microscopy to identify the presence of PA invasion. Invasiveness via histology was defined as the infiltration and/or destruction of basal sellar dura cells by a PA, as observed in the microscopic field of vision.

### 2.5. Classification of PA Invasiveness

As shown in Table 1, Invasiveness Classification 1 involves the categorization of PAs into 4 grades according to radiological, surgical, and histological indicators as follows: grade 0, non-invasive PAs; grade I, PAs meeting one of three criteria for invasive PAs; grade II, PAs that meet any two of three criteria; grade III, PAs meeting all three criteria.

Invasiveness Classification 2 includes Knosp-classification- and surgical-invasiveness-based parameters, where PAs were divided into the following 5 grades: grade 0, Knosp grade 0 without surgical invasiveness; grade 1, Knosp grade 1–2 without surgical invasiveness; grade 2, Knosp grade 0–2 with surgical invasiveness, or Knosp grade 3, PAs without surgical invasiveness; grade 3, Knosp grade 3 PAs with surgical invasiveness; grade 4, Knosp grade 4. Based on the scheme, grade 3–4 PAs were considered invasive, grade 0–1 PAs were considered non-invasive, and grade 2 referred to radiologically invasive PAs without surgical invasiveness or surgically invasive PAs without radiological invasiveness. All PAs were classified by the same author (L.L.).

### 2.6. Statistical Analysis

Categorical data are presented as percentages, and continuous variables are presented as means ± standard deviation (SD). The analysis of variance was used to compare continuous variables, while Chi-square tests were used to compare categorical variables. Kaplan–Meier (KM) analysis and the log-rank test were used to estimate RPFS. Receiver-operating characteristic (ROC) curves were used to determine which classification scheme best predicted GTR and RPFS. R software (version 3.6.3) was used to perform statistical analysis. Data were visualized using “ggplot” R packages (version 3.3.3). Values of *p* < 0.05 were considered statistically significant.

## 3. Results

### 3.1. Baseline Characteristics

A total of 903 patients with PAs who underwent endoscopic-assisted micro resection for primary TTS were included in this study (Table 2). The mean age of included patients was 48.46 years, and 396 (43.9%) were female. The mean duration of follow-up was 51.5 months (range: 10.3–107.6 months). Among the PAs analyzed, the following clinical subtypes were observed: nonfunctioning, 489 patients (54.2%); PRL-secreting, 195 patients (21.6%); GH-secreting, 127 patients (14.1%); ACTH-secreting, 34 patients (3.8%); TSH-secreting, 4 patients (0.4%), and plurihormonal, 54 patients (6.0%). The mean maximum PA diameter was 22.90 mm. Among all cases considered, 136 (16.2%) had PAs with Ki-67 scores ≥ 3, and 335 (37.1%), 542 (60%), and 481 (53.3%) had PAs displaying radiographic, histological, and surgical features of invasiveness, respectively. Distributions of features of PA invasiveness identified among the study population are shown in Figure 2.

### 3.2. Characteristics of Invasiveness Classification 1

All PAs were classified based on Invasiveness Classification 1 grade (Table 1). Based on Invasiveness Classification 1, 23.5% (*n* = 212) of patients had non-invasive PAs, 27.4% (*n* = 247) had grade I, 24.5% (*n* = 221) had grade II, and 24.7% (*n* = 223) had grade III. Among the four grades considered, age (*p* < 0.001), clinical subtype (*p* < 0.001), maximum PA diameter (*p* < 0.001), and degree of invasiveness (Table 2) significantly differed. Patients with PAs of higher grade tended to be younger and have a larger maximum PA diameter than those with lower-grade PAs. PA grade distributions varied based on clinical subtype (Figure 3). Greater proportions of non-invasive and low-grade PA invasiveness were observed among ACTH- and PRL-secreting PAs versus other subtypes.

### 3.3. Surgical Outcomes and Recurrence/Progression

As shown in Table 3, the higher the grade of invasiveness, the lower the GTR rate, and the higher the likelihood of recurrence/progression. Knosp classification tended to be consistent with Invasiveness Classification 1. In particular, GTR and recurrence/progression rates of Knosp 3A and Knosp 3B PAs were 62.5% and 43.1%, and 46.2% and 67.7%, respectively.

Among the three methods for assessing invasiveness considered, radiological evaluations identified PAs with the lowest GTR rates (48.7%), followed by surgical and histological methods (58.8% and 71.4%, respectively). Correspondingly, the recurrence/progression rates of PAs with radiological indicators of invasiveness were the highest (59.7%), followed by those with surgical and histological invasiveness indicators (50.5% and 35.4%, respectively). In particular, GTR rates of PAs with CSI were significantly less than those of PAs without CSI, whereas recurrence/progression rates were greater.

RPFS of PAs of different Invasiveness Classification grades are presented as KM curves (Figure 4). The higher the grade, the shorter the RPFS. Further, RPFS values of PAs of different grades significantly differed. Moreover, KM curves for radiological (hazard ratio (HR) 5.11, 95% confidence interval (CI) 3.98–6.57, *p* < 0.001), surgical (HR 6.40, 95% CI: 5.09–8.06, *p* < 0.001) and histological (HR 1.44, 95% CI: 1.14–1.81, *p* = 0.003) predictors of PA invasiveness revealed that each effectively predicted postoperative PA prognosis; however, the effect of histological assessment was less than that of radiological and surgical.

### 3.4. Knosp Grade 2–3 PAs with and without Surgical Invasiveness

As shown in Figure 5, among patients with PAs of Knosp grade 2, the mean GTR rate was 82.9%. When Knosp grade 2 was further classified based on the presence or absence of surgical indicators of invasiveness, GTR rates of the subgroups significantly differed, with GTR rates of PAs of Knosp grade 2, with and without surgical invasiveness, being 71.7% vs. 93.1%, respectively (*p* = 0.003). Similarly, Knosp grade 3 was stratified based on the presence or absence of surgical invasiveness indicators. The mean GTR rate of Knosp grade 3 PAs was 57.0%. The GTR rates of Knosp grade 3 Pas, with and without surgical invasiveness indicators, significantly differed (53.4% vs. 81.3%, respectively; *p* = 0.003). Based on KM curves (Figure 6), Knosp grade 2 (HR 4.63, 95% CI: 2.13–10.06, *p* < 0.001) and Knosp grade 3 (HR 2.23, 95% CI: 1.39–3.59, *p* = 0.011) PAs with surgical indicators of invasiveness predicted prognosis significantly better than their corresponding Knosp grades without surgical invasiveness.

### 3.5. Invasiveness Classification 2

Invasiveness Classification 2, which depends on a combination of radiographic and surgical characteristics, is presented in Table 1. As shown in Table 3, the higher the grade of invasiveness, the lower the GTR rate, and the more likely recurrence/progression is. GTR and recurrence/progression rates of invasive (grade 3–4) and non-invasive (grade 0–1) PAs in Invasiveness Classification 2 were 62.5% and 43.1%, and 46.2% and 67.7%, respectively. RPFS of PAs of Invasiveness Classification 2 are presented as KM curves (Figure 7). RPFS values of invasive (grade 3–4) PAs were shortest, followed by grade 2 and non-invasive (grade 0–1) PAs. Moreover, KM curves for two subgroups in grade 2 revealed that poor prognosis of intermediate grade was not well predicted by radiological and surgical indicators of invasiveness alone.

### 3.6. ROC Curves of Classification Methods

ROC curves for predicting surgical outcome and recurrence/progression status, using Knosp classification, Invasiveness Classification 1, and Invasiveness Classification 2, are shown in Figure 8. We found that both Invasiveness Classification 2 and revised-Knosp classification could accurately predict GTR (area under the curve (AUC) 0.853 and AUC 0.847, respectively; *p* = 0.353), which were better than Invasiveness Classification 1 (AUC 0.800; *p* < 0.001 and *p* < 0.001, respectively). Furthermore, Invasiveness Classification 2 most accurately predicted RPFS (AUC 0.812), followed by revised-Knosp classification, which was also an accurate predictor (AUC 0.795, *p* = 0.044). Invasiveness Classification 1 was the poorest predictor of RPFS (AUC 0.758; *p* < 0.001 and *p* = 0.002, respectively).

## 4. Discussion

PA invasiveness is a well-known prognostic predictor of disease-free status following surgical PA removal [4,19]. In the present study, PA invasiveness was assessed and classified based on the radiological, surgical, and histological features of 903 patients who were surgically treated at our institution. Moreover, Invasiveness Classification 1 was established to predict invasiveness based on all three perspectives considered. Furthermore, we demonstrated that radiological and surgical indicators of invasiveness more accurately predicted GTR and RPFS rates than histological indicators did. Further, Invasiveness Classification 2, which comprised the Knosp classification and surgical invasiveness grade, predicted RPFS more accurately than Invasiveness Classification 1 and Knosp classification alone. These findings improve our understanding of PA invasiveness, and have the potential to inform clinicians, regarding the diagnosis and treatment of invasive PAs.

### 4.1. Radiological Invasiveness

PAs are typically considered invasive when they are classified as Knosp grade 3–4 on MRI [3,10,20]. Even though the Knosp classification system was designed to predict CSI during surgery, rather than invasion per se, it has been extensively studied and is widely used because it is an accurate predictor of prognosis [3,10,21,22,23]. Araujo-Castro et al. demonstrated that the surgical cure rate decreased as Knosp grade (*p* < 0.001) increased [3]. Regarding the prediction of a surgical cure, Knosp classification (AUC: 0.820) is more accurate than any other radiological classification method, including Hardy–Wilson (AUC 0.654) [3]. This may be due to the capacity of the Knosp classification to describe CSI. Several studies have shown that CSI is the factor that most significantly and independently affects surgical outcome [3,5,17,24,25]. However, rates of radiologically predicted and actual CSI differ, particularly because invasion and compression are difficult to distinguish using radiological images [2,3,26]. Dhandapani et al. confirmed that the rate of false-positives for Knosp grade 3 is very high when evaluating an endoscopic series [12], and improved methods for identifying parasellar invasion are needed.

This was also the view of the Knosp team, who further classified Knosp grade 3 into grade 3A and 3B subtypes, based on clinical features and prognostic differences [17]. The researchers described the importance of distinguishing Knosp grades 3A, 3B, and 4 when assessing parasellar invasion identified during endoscopic surgery [17,27]. In this study, the rates of Knosp grades 3A, 3B and 4 were 65.8%. 73.8% and 100%, respectively. Additionally, 10% of Knosp grade 2 PAs were identified as invasive via an endoscopic series, despite the fact that they were generally considered non-invasive via Knosp classification [17]. Outcomes of treatment of Knosp 3A and Knosp 3B PAs significantly differed [3,17,27]. The GTR rates of patients with Knosp 3B PAs were lower than those with Knosp 3A (43.1% vs. 62.5%, respectively; *p* = 0.006). Further, the rate of recurrence/progression in patients with Knosp 3B versus 3A were elevated (67.7% versus 46.2%, respectively; *p* = 0.003). Work by Araujo also revealed that patients with Knosp 3B PAs were less likely to be cured via surgery than those with Knosp 3A PA (30.0% versus 56.0%, respectively; *p* = 0.164) [3]. Knosp grade 3A PAs tended to be more similar to those classified as Knosp grade 2, whereas the behavior of Knosp grade 3B PAs more closely mirrored those of Knosp grade 4 [3].

### 4.2. Surgical Invasiveness

PAs extending into the cavernous sinus are generally considered invasive from a radiological perspective but not necessarily from surgical or histological points of view [13,17]. As surgical visibility and parasellar structure identification are improved via endoscopy, surgeons will be increasingly able to view the MWCS directly, collect more data, and distinguish parasellar area compression from biological invasion [16]. Frank et al. found that the MWCS is frequently displaced without invasion in Knosp 3A PAs [28]. Some studies have also reported defects in the cavernous sinus wall or thinning of its parts, which may explain the extension of PAs to the parasellar area [29,30,31,32]. These types of PAs may be identified as invasive via radiology, despite the fact that they may not be biologically invasive [30,33].

Several studies have associated MWCS involvement with Knosp grade [17,34,35]. In fact, differences between Knosp grade correlations with invasiveness were observed intraoperatively, underscoring the fact that radiological features are imperfect predictors of PA invasiveness [3,17,36]. Invasiveness, which implies that invasive growth of PAs to surrounding structures occurs, is different from the lateral shift of CS structure, which is observed after simple tumor extension [17]. Surgical examination remains the most effective means for distinguishing invasion from compression [17], and functions as a reliable standard for the identification of CSI [12,37].

In this study, surgical CSI occurred in 36.2% of cases, a value similar to that of one prior report (37%) [31], but higher than that of another (16%) [17], which excluded cases exclusively involving MWCS invasion [17]. Of the 335 PAs radiologically identified as invasive, 92 (27.5%) did not invade the cavernous sinus or MWCS, and instead compressed the MWCS. These PAs without intraoperative CSI were easier to remove than radiologically invasive PAs with intraoperative CSI (GTR: 60.9% versus 44.0%). CSI provided the indications for primary Gamma Knife radiosurgery, and Palmisciano et al. found that primary Gamma Knife radiosurgery has favorable efficacy in large tumors infiltrating the cavernous sinuses [38]. In this study, in addition to CSI, invasion also included visible destruction of sphenoid sinus, clivus, and sellar diaphragm within the surgical visual field. The overall rate of surgical invasiveness was 53.3%, which was higher than that of radiological invasiveness (37.1%). Although the rate of invasiveness was higher, the predictive value of surgical invasiveness (HR 6.40, 95% CI: 5.09–8.06, *p* < 0.001) for RPFS was better than that of radiological invasiveness (HR 5.11, 95% CI 3.98–6.57, *p* < 0.001), highlighting the importance of observations made during an operation.

### 4.3. Histological Invasiveness

The histopathological evaluation of PAs invading the basal sellar dura allows us to better understand the biological behavior of the tumors [13]. Although histopathological analysis of the MWCS is the gold standard for the diagnosis of CSI, routine biopsies are too dangerous [17]. Micko et al. stated that direct connective tissue biopsy of the MWCS is not feasible as a routine practice due to the risk of neurovascular structural damage, even in situations in which endoscopic techniques are used [17]. Moreover, Yamada et al. revealed that immunohistochemical results of samples obtained from different parts of a PA during surgery could be considered representative of entire adenomas [39]. Meij et al. also suggested that invasiveness assessed via dural biopsy sampling satisfactorily approximated true PA invasiveness relative to adjacent tissues [13]. This provides a theoretical basis for using the histological features of basal sellar dura samples from PAs to represent whole adenomas.

Histological invasiveness rates of basal sellar dura samples have reported to range from 43% to 85% (60.0% in this study) [13,17,40]. More than 50% of patients with nonfunctioning PAs and 30% to 35% of patients with active, endocrine PAs have tumors with dural invasion [13]. Surgical or radiological findings are better predictors of invasiveness than histological findings [41]; therefore, surgical or radiological invasiveness findings affect decision making to a greater extent than does dura mater infiltration isolated via microscopy [40]. This may be due to the fact that histological invasiveness assessment is strictly dependent on technical progress, questions regarding the representativeness of tissue samples, and the lack of recognized and effective biomarkers [13,40]. Quantification of the Ki-67 labeling index can distinguish pituitary carcinomas (11.9 ± 3.4% on average) and other adenomas (1.4 ± 0.15% on average) [42]. Nonetheless, whether Ki-67 is an efficient prognostic factor for PAs remains controversial [43]. In a review involving 28 studies on Ki-67, 18 studies reported high Ki-67 expression in recurrent adenomas, while the other 10 studies showed no correlation [44]. Although the expression level in Ki67 bears no relationship to invasiveness in our study (*p* = 0.455), we still pay high attention to patients with high Ki67 in clinical practice, and remind them that regular follow-up is required after treatment. In the present study, even though histological invasiveness could effectively predict RPFS (HR 1.44, 95% CI: 1.14–1.81, *p* = 0.003), its prognostic value was less than that of radiological (HR 5.11, 95% CI 3.98–6.57, *p* < 0.001) and surgical invasiveness (HR 6.40, 95% CI: 5.09–8.06, *p* < 0.001).

Despite its limitations, histological assessment provides cellular information regarding invasiveness, and further clarifies the biological behavior of PAs. Dickerman et al. believed that occult dural invasion is of clinical importance, since it provides a basis for tumor recurrence or persistence and endocrine diseases [45]. The ability of PAs to invade the dura mater may be reflected in their tendency to persist or relapse after TTS, which may affect remission rates of endocrine-active PAs and nonfunctioning PAs [40,41]. PA tissue may also persist due to dural invasion in patients with GTR [13]. Meij et al. believed that recurrence rates (after confirming the first remission) are not consistently associated with dural invasion, and that the primary significance of dural invasion is the presence of residual tumor tissue [13]. PAs with histological invasiveness need to be vigilant against the histological vascular alterations; Fraioli et al. showed that 71.1% (32/45) of hemorrhagic PAs were found to be invasive, suggesting that the invasiveness of PAs may lead to hemorrhagic complications in pathological findings [46]. Attention should be paid to the identification of other sellar tumors that are also invasive. For example, Rahman et al. introduced primary sellar neuroblastoma mimicking invasive pituitary adenoma [47]. In general, the role of histological invasiveness in GTR and recurrence rate prediction in PAs is controversial; however, occult dural invasion, which provides important supplemental micro-level information, is of clinical value. Since sampling can be performed safely and conveniently, it has been suggested that routine histological evaluation of the basal sellar dura may be performed. When further confirmation of CSI is needed, MWCS samples can be carefully obtained by experienced surgeons for histopathological evaluation. Accurate data regarding CSI status have the potential to inform the individual diagnosis and treatment of PAs with CSI.

### 4.4. Invasiveness Classification

In the present study, the Invasiveness Classification 1 scheme was established based on radiological, surgical and histological methods for evaluating PA invasiveness. As the invasiveness grade increased, rates of GTR decreased and recurrence/progression increased. Recurrence/progression rates of non-invasive PAs were 3.8% upon 9-year follow-up, which was significantly lower than that of radiologically non-invasive PAs (16.0%, *p* < 0.001). Moreover, significant differences in RPFS rates were observed among PAs with different invasive grades (Figure 4A). AUC values for Invasiveness Classification 1, when predicting GTR and RPFS, were lower than those of the Knosp classification. This finding may have been due to the large proportion of PAs with histological invasiveness identified, which improved the sensitivity by which invasion was differentiated but diminished the effectiveness of surgical outcome and prognosis prediction. 

Our study showed that identification of Knosp grade 2–3 PA identification can be stratified based on the presence or absence of surgical invasiveness. The GTR rate of Knosp 2 PAs with surgical invasiveness was significantly lower than that of Knosp 2 PAs without surgical invasiveness (71.7% versus 93.1%, respectively; *p* = 0.003). Likewise, the GTR rate of Knosp 3 PAs with surgical invasiveness was significantly lower than that of Knosp 3 grade PAs without surgical invasiveness (53.4% versus 81.3%, respectively; *p* = 0.003). Knosp grade 2 (HR 4.63, 95% CI: 2.13–10.06, *p* < 0.001) and Knosp grade 3 (HR 2.23, 95% CI: 1.39–3.59, *p* = 0.011) PAs with surgical invasiveness also better predicted RPFS versus corresponding Knosp grades without surgical invasiveness. Therefore, the Knosp classification was combined with surgical invasiveness to establish Invasive Classification 2 (Table 1), which effectively revises the Knosp classification using surgical invasiveness data to improve the prediction of invasive PAs. Based on this scheme, grade 3–4 PAs were considered invasive, grade 0–1 PAs were considered non-invasive, and grade 2 referred to unilaterally invasive PAs, which included radiologically invasive PAs without surgical invasiveness or surgically invasive PAs without radiological invasiveness. RPFS values of invasive (grade 3–4) PAs were shortest, followed by grade 2 and non-invasive (grade 0–1) PAs. Furthermore, KM curves for two subgroups in grade 2 revealed that poor prognosis of the intermediate grade was not well predicted by radiological and surgical indicators of invasiveness alone.

Invasiveness Classification 2 most accurately predicted GTR (AUC 0.853), followed by Knosp classification (AUC 0.847), and Invasiveness Classification 1 (AUC 0.800). Likewise, when predicting RPFS, Invasiveness Classification 2 (AUC 0.812) was significantly more reliable than Knosp classification (AUC 0.795) and Invasiveness Classification 1 (AUC 0.758). Invasiveness Classification 2 most accurately predicted RPFS (AUC 0.812), followed by revised-Knosp classification, which was also an accurate predictor (AUC 0.795, *p* = 0.044). Invasiveness Classification 1 was the poorest predictor of RPFS (AUC 0.758; *p* < 0.001 and *p* = 0.002, respectively), indicating that a combination of radiological and surgical information improves the identification of invasive characteristics of PAs, PA behavior prediction, and surgical outcome and prognosis prediction in PA patients.

### 4.5. Limitations

This study has several limitations. First, this was a single-center retrospective analysis, within which a selection bias was inherent. Second, the classification of invasive behavior based on radiological, surgical, and histological findings requires validation using data from patients treated at other centers, preferably via a prospectively designed study. Moreover, our definition of invasiveness, especially histological invasiveness, may overestimate the presence of invasive PAs. Notably, other centers may not collect basal sellar dura samples for histological evaluation; therefore, additional assessment using data from patients treated at other centers may be difficult.

## 5. Conclusions

Invasiveness Classification 1 was established on the basis of radiological, surgical, and histological features, and the scheme could recognize non-invasive PAs with long-term RPFS. Radiological and surgical indicators of PA invasiveness were more accurate predictors of GTR and RPFS than histological indicators were. A scheme named Invasiveness Classification 2, which better predicted RPFS than Knosp classification and Invasiveness Classification 1, was established by combining Knosp classification and surgical criteria. We believe that the convenience and good predictive power of Invasiveness Classification 2 will facilitate the identification of invasive PAs and the development of improved treatment plans.

## Figures and Tables

**Figure 1 jcm-11-02464-f001:**
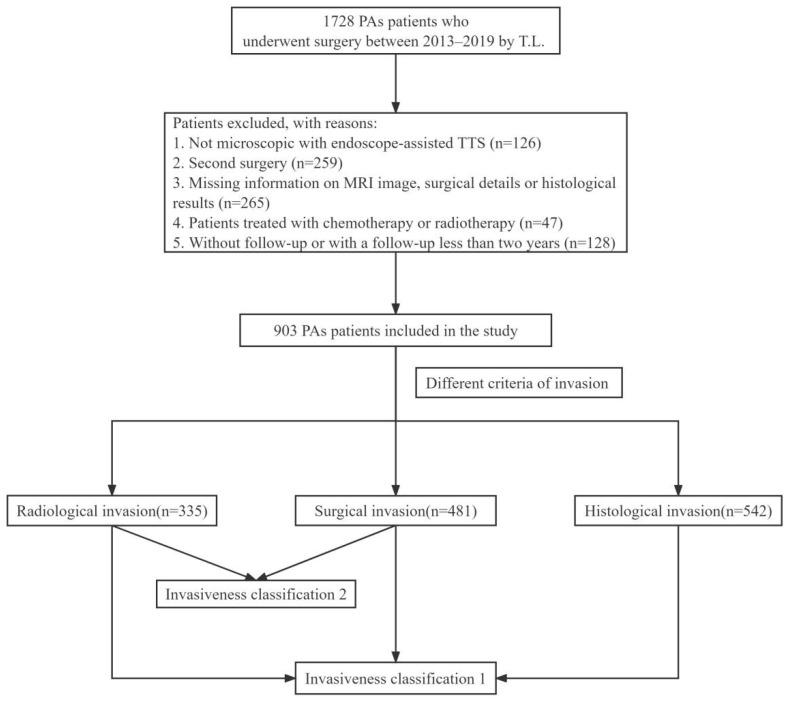
Flowchart summarizing the enrollment strategy and design of the study. PAs: Pituitary Adenomas; TTS: Transnasal-Transsphenoidal Surgery.

**Figure 2 jcm-11-02464-f002:**
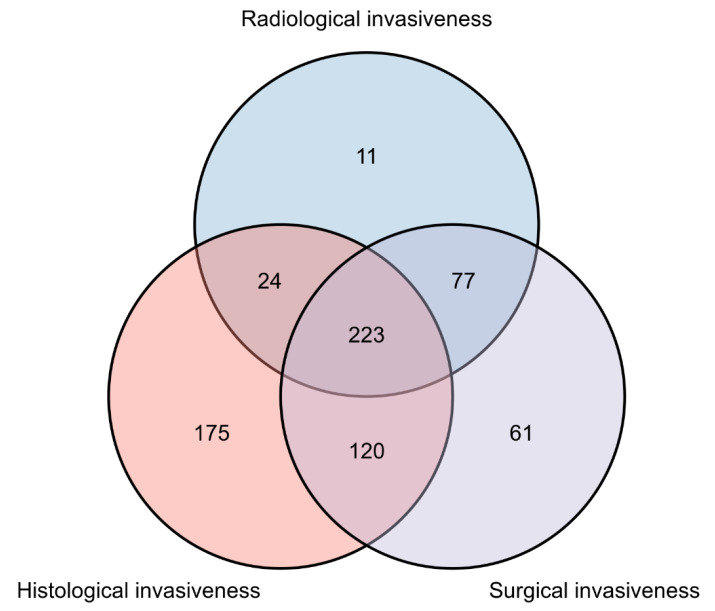
Venn chart illustrating the number and interactive information of patients with radiological, surgical, and histological invasiveness.

**Figure 3 jcm-11-02464-f003:**
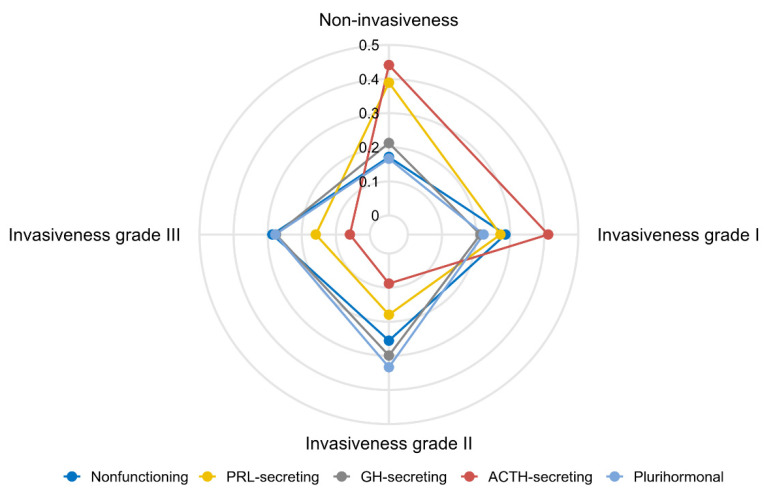
Radar chart illustrating the proportional distribution of invasiveness for different clinical subtypes. PRL: Prolactin; GH: Growth Hormone; ACTH: Adrenocorticotropic Hormone.

**Figure 4 jcm-11-02464-f004:**
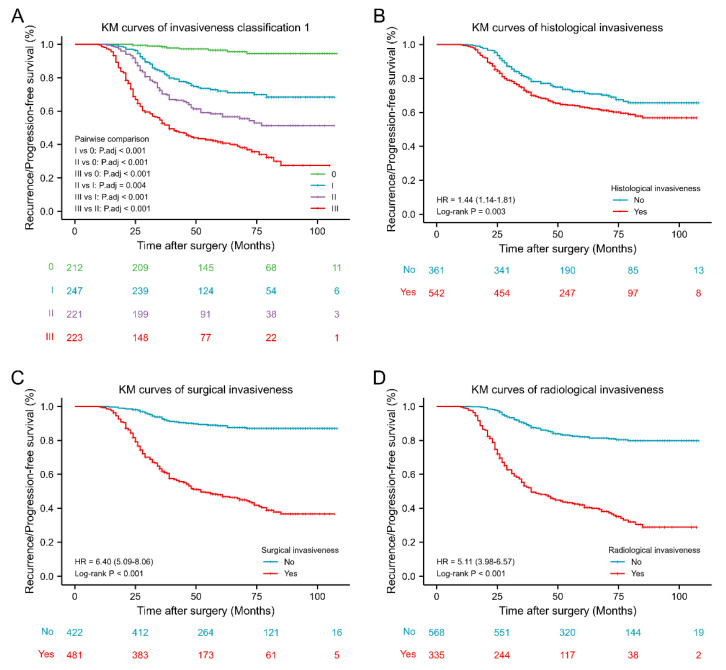
Kaplan–Meier survival curves of different invasiveness groups. (**A**) Kaplan–Meier survival curves of invasiveness classification 1. (**B**) Kaplan–Meier survival curves of histological invasiveness. (**C**) Kaplan–Meier survival curves of surgical invasiveness. (**D**) Kaplan–Meier survival curves of radiological invasiveness. KM: Kaplan–Meier; HR: Hazard Ratio.

**Figure 5 jcm-11-02464-f005:**
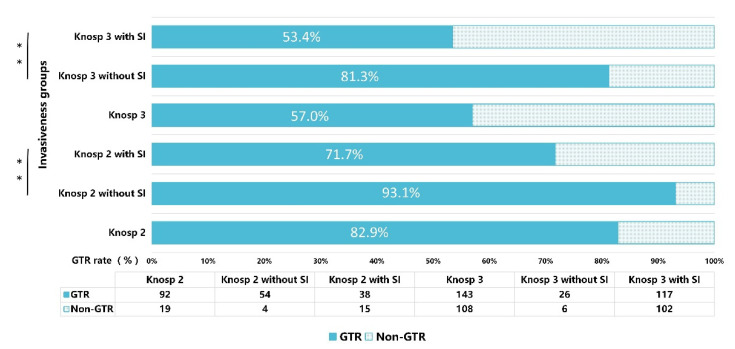
Bar chart illustrating the gross-total resection (GTR) rates of Knosp grade 2–3 with or without surgical invasiveness. SI: surgical invasiveness. ** *p* < 0.01.

**Figure 6 jcm-11-02464-f006:**
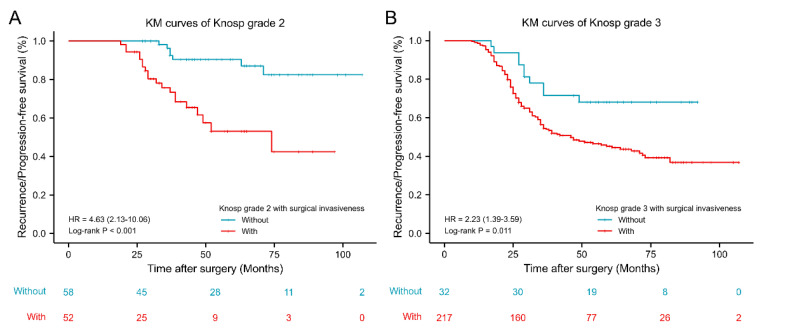
Kaplan–Meier survival curves of Knosp grade 2–3 with or without surgical invasiveness. (**A**) Kaplan–Meier survival curves of Knosp grade 2 with or without surgical invasiveness. (**B**) Kaplan–Meier survival curves of Knosp grade 3 with or without surgical invasiveness. KM: Kaplan–Meier; HR, Hazard Ratio.

**Figure 7 jcm-11-02464-f007:**
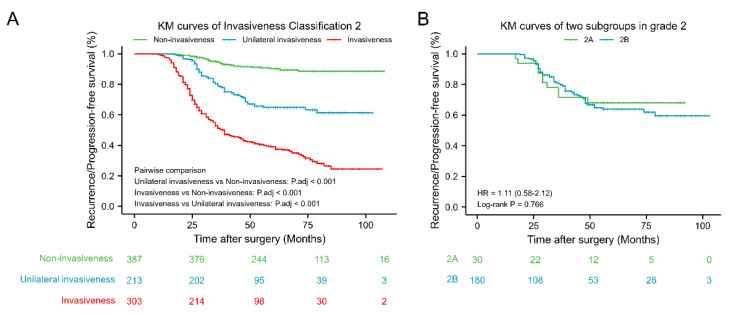
Kaplan–Meier survival analysis of Invasion Classification 2. (**A**) Kaplan–Meier survival curves of Invasion Classification 2. (**B**) Kaplan–Meier survival curves of two subgroups in grade 2. Non-invasiveness: Invasion Classification grade 0–1; Unilateral invasiveness: Invasion Classification grade 2; Invasiveness: Invasion Classification grade 3–4; 2A: Knosp grade 3 PAs without surgical invasiveness; 2B: Knosp grade 0–2 PAs with surgical invasiveness. KM: Kaplan–Meier; HR, Hazard Ratio.

**Figure 8 jcm-11-02464-f008:**
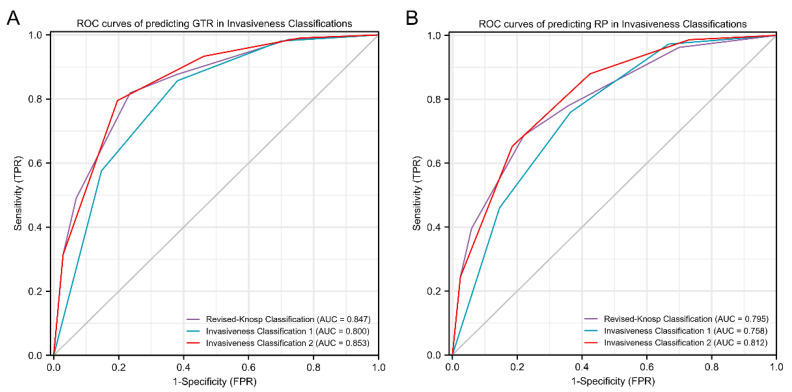
Receiver operator characteristic (ROC) analysis of gross-total resection (GTR) and recurrence/progression (RP). ROC curves of predicting GTR in revised-Knosp classification, Invasiveness Classification 1, and invasiveness classification 2 (**A**). ROC curves of predicting RP for revised-Knosp classification, Invasiveness Classification 1, and Invasiveness Classification 2 (**B**). CI: Confidence Interval; FPR: False Positive Rate; TPR: True Positive Rate.

**Table 1 jcm-11-02464-t001:** Invasiveness classifications of PAs.

	Different Criteria for Invasiveness
Radiological invasiveness	The Knosp grade 3–4
Surgical invasiveness	The invasive and destructive growth of cavernous sinus, sphenoid sinus, ramp and diaphragm sella under direct vision during surgery.
Histological invasiveness	The infiltration and/or destruction of the basal sellar dura samples by PAs in the microscopic field of vision.
	**Invasiveness classification 1**
Grade 0	PAs that meet none of three invasive criteria
Grade I	PAs that meet only one of three invasive criteria
Grade II	PAs that meet any two of three invasive criteria
Grade III	PAs that meet all three invasive criteria
	**Invasiveness classification 2**
Grade 0	Knosp grade 0 PAs without surgical invasiveness
Grade 1	Knosp grade 1–2 PAs without surgical invasiveness
Grade 2	Knosp grade 0–2 PAs with surgical invasiveness, or Knosp grade 3 PAs without surgical invasiveness
Grade 3	Knosp grade 3 PAs with surgical invasiveness
Grade 4	Knosp grade 4 PAs

PAs, Pituitary Adenomas.

**Table 2 jcm-11-02464-t002:** Characteristics of different invasiveness grades (Invasiveness Classification 1).

Characteristic	Total (*n* = 903)	Non-Invasiveness (*n* = 212)	Invasiveness Grade I (*n* = 247)	Invasiveness Grade II (*n* = 221)	Invasiveness Grade III (*n* = 223)	Method	*p*-Value
Age, year	48.46 ± 11.38	49.52 ± 11.27	48.82 ± 11.56	48.18 ± 10.83	47.35 ± 12.23	ANOVA	0.404
Gender						Chisq.test	0.952
Female	396 (43.9%)	94 (44.3%)	106 (42.9%)	100 (45.2%)	96 (43.1%)		
Male	507 (56.1%)	118 (55.7%)	141 (57.1%)	121 (54.8%)	127 (56.9%)		
Clinical subtypes						Chisq.test	<0.001
Nonfunctioning	489 (54.2%)	84 (39.6%)	140 (56.7%)	125 (56.6%)	140 (62.8%)		
PRL-secreting	195 (21.6%)	76 (35.8%)	53 (21.5%)	35 (15.8%)	31 (13.9%)		
GH-secreting	127 (14.1%)	27 (12.7%)	27 (10.9%)	38 (17.2%)	35 (15.7%)		
ACTH-secreting	34 (3.8%)	15 (7.1%)	14 (5.7%)	3 (1.4%)	2 (0.9%)		
TSH-secreting	4 (0.4%)	1 (0.5%)	1 (0.4%)	2 (0.9%)	0 (0.0%)		
Plurihormonal	54 (6.0%)	9 (4.2%)	12 (4.9%)	18 (8.1%)	15 (6.7%)		
Maximum PA diameter, mm	22.90 ± 12.18	15.19 ± 8.27	20.77 ± 9.39	24.77 ± 11.17	30.74 ± 14.36	ANOVA	<0.001
Knosp grade						Chisq.test	<0.001
0	195 (21.6%)	114 (53.6%)	69 (27.9%)	12 (5.4%)	0 (0.0%)		
1	262 (29.0%)	78 (36.8%)	112 (45.3%)	72 (32.6%)	0 (0.0%)		
2	111 (12.3%)	20 (9.4%)	55 (22.3%)	36 (16.3%)	0 (0.0%)		
3A	184 (20.4%)	0 (0.0%)	11 (4.5%)	69 (31.2%)	104 (46.6%)		
3B	65 (7.2%)	0 (0.0%)	0 (0.0%)	6 (2.7%)	59 (26.5%)		
4	86 (9.5%)	0 (0.0%)	0 (0.0%)	26 (11.8%)	60 (26.9%)		
Radiological Invasiveness						Chisq.test	<0.001
No	568 (62.9%)	212 (100.0%)	236 (95.5%)	120 (54.3%)	0 (0.0%)		
Yes	335 (37.1%)	0 (0.0%)	11 (4.5%)	101 (45.7%)	223 (100.0%)		
Surgical Invasiveness						Chisq.test	<0.001
No	422 (46.7%)	212 (100.0%)	186 (75.3%)	24 (10.9%)	0 (0.0%)		
Yes	481 (53.3%)	0 (0.0%)	61 (24.7%)	197 (89.1%)	223 (100.0%)		
Histological Invasiveness						Chisq.test	<0.001
No	361 (40.0%)	212 (100.0%)	72 (29.1%)	77 (34.8%)	0 (0.0%)		
Yes	542 (60.0%)	0 (0.0%)	175 (70.9%)	144 (65.2%)	223 (100.0%)		
Cavernous Sinus Invasion						Chisq.test	<0.001
No	576 (63.8%)	212 (100.0%)	217 (87.9%)	100 (45.2%)	47 (21.1%)		
Yes	327 (36.2%)	0 (0.0%)	30 (12.1%)	121 (54.8%)	176 (78.9%)		
Sphenoid Sinus Invasion						Chisq.test	<0.001
No	616 (68.2%)	212 (100.0%)	200 (81.0%)	115 (52.0%)	89 (39.9%)		
Yes	287 (31.8%)	0 (0.0%)	47 (19.0%)	106 (48.0%)	134 (60.1%)		
Ki-67 ≥ 3 (*n* = 840)						Chisq.test	0.455
No	704 (83.8%)	159 (81.1%)	196 (86.7%)	169 (82.8%)	180 (84.1%)		
Yes	136 (16.2%)	37 (18.9%)	30 (13.3%)	35 (17.2%)	34 (15.9%)		

PRL: Prolactin; GH: Growth Hormone; ACTH: Adrenocorticotropic Hormone; TSH: Thyroid-Stimulating Hormone; PA: Pituitary Adenoma; ANOVA: Analysis of Variance; Chisq.test: Chi-square test.

**Table 3 jcm-11-02464-t003:** Surgical results and recurrence/progression status of different invasiveness groups.

Variable	GTR			Recurrence/Progression		
	*n* (%)	OR, 95% CI	*p*-Value	*n* (%)	HR, 95% CI	*p*-Value
Classification 1 grade						
0	208 (98.1%)	1.000		8 (3.8%)	1.00	
I	216 (87.4%)	0.163 [0.056–0.476]	0.001	62 (25.1%)	7.85 [4.91–12.53]	<0.001
II	151 (68.3%)	0.053 [0.019–0.148]	<0.001	87 (39.4%)	13.48 [8.99–20.20]	<0.001
III	102 (45.7%)	0.016 [0.006–0.045]	<0.001	134 (60.1%)	23.30 [16.69–32.52]	<0.001
Revised-Knosp grade						
0	189 (96.9%)	1.000		11 (5.6%)	1.00	
1	233 (88.9%)	0.162 [0.048–0.549]	0.003	53 (20.2%)	4.31 [2.64–7.05]	<0.001
2	92 (82.9%)	0.129 [0.036–0.467]	0.002	27 (24.3%)	5.27 [2.66–10.43]	<0.001
3A	115 (62.5%)	0.026 [0.008–0.085]	<0.001	85 (46.2%)	10.78 [7.18–16.16]	<0.001
3B	28 (43.1%)	0.012 [0.003–0.041]	<0.001	44 67.7%)	19.63 [9.70–39.71]	<0.001
4	20 (23.3%)	0.005 [0.001–0.016]	<0.001	71 (82.6%)	26.28 [15.25–45.27]	<0.001
Classification 2 grade						
0	165 (97.6%)	1.000		4 (2.4%)	1.00	
1	199 (91.3%)	0.254 [0.085–0.761]	0.014	31 (14.2%)	6.96 [3.59–13.52]	<0.001
2	177 (83.1%)	0.119 [0.042–0.342]	<0.001	66 (31.0%)	16.42 [10.28–26.23]	<0.001
3	116 (53.5%)	0.028 [0.010–0.078]	<0.001	119 (54.8%)	32.89 [23.09–46.86]	<0.001
4	20 (23.3%)	0.007 [0.002–0.022]	<0.001	71 (82.6%)	62.24 [36.10–107.29]	<0.001
Radiological Invasiveness						
No	514 (90.5%)	1.000		91 (16.0%)	1.00	
Yes	163 (48.7%)	0.100 [0.070–0.142]	<0.001	200 (59.7%)	5.11 [3.98–6.57]	<0.001
Surgical Invasiveness						
No	394 (93.4%)	1.000		48 (11.4%)	1.00	
Yes	283 (58.8%)	0.102 [0.066–0.155]	<0.001	243 (50.5%)	6.40 [5.09–8.06]	<0.001
Histological Invasiveness						
No	290 (80.3%)	1.000		99 (27.4%)	1.00	
Yes	387 (71.4%)	0.611 [0.444–0.841]	0.003	192 (35.4%)	1.44 [1.14–1.81]	0.003
Cavernous Sinus Invasion						
No	503 (87.3%)	1.000		106 (18.4%)	1.00	
Yes	174 (53.2%)	0.165 [0.119–0.229]	<0.001	185 (56.6%)	4.06 [3.16–5.22]	<0.001

GTR: Gross-Total Resection; OR: Odds Ratio; HR: Hazard Ratio; CI: Confidence Interval.

## Data Availability

The data that support the findings of this study are included in the article. Further inquiries are available from the corresponding author upon reasonable request.

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
