# Peer review of "Classifying Pituitary Adenoma Invasiveness Based on Radiological, Surgical and Histological Features: A Retrospective Assessment of 903 Cases"

_jcm, 2022, doi:10.3390/jcm11092464_

Round 1

Reviewer 1 Report

This retrospective study compares the predictive value of two invasiveness-based classifications on the outcome of surgery for pituitary adenomas. Classification 1 (C1) is based on radiological, surgical and histological assessment of invasiveness. Classification 2 (C2) combines  a well-established radiological classification (Knosp) with invasiveness verified at the time of surgery. The data are derived from 905 patients followed for a mean of 4 years with gross tumour removal (GTR) and tumour progression/recurrence as outcome measures.

They report C2 to be superior to C1 and that surgical verification of tumour invasion to be a dominant factor in both models. They observed that a revised Knosp (radiological) classification performed as well as C2 in predicting GTR and progression free survival. Histological invasion itself had little influence on prognosis.

The authors conclude that radiological and surgical invasiveness are valuable clinical prognostic indicators and that C2 based classification may be useful in developing a treatment plan for pituitary adenomas.

This is an interesting, carefully written, thoroughly analysed and well-presented study of clinical relevance. The following comments are offered to clarify certain points and to strengthen the paper:

  1. Statistical analysis
  2. Please clarify whether t-test comparisons are corrected for the number of groups.
  3. In Table 1 explain what comparisons are represented by the p values.
  4. Is the p value for age (<0.001) correct for the comparions between the invasiveness groups, the ages of which are similar.
  5. ROC comparison between C2 and Revised Knosp (Section 3.6): The differences in AUCs for GTR and RPFS between the two are very small. Unless the AUCs are statistically significant, the authors cannot claim superiority of one classification over the other (see below) or over C1.

  1. Discussion
  2. From this reviewer’s assessment, the Revised Knosp classification appears to perform as well as C2. This should be discussed including the important implication that the former is a reliable surrogate marker of C2 and immensely useful for predicting prognosis, obviating the need to verify surgical invasion.
  3. Much as been written about Ki67, the expression level of which, bears no relationship to invasiveness or outcome. The usefulness of Ki67 should be discussed.

Both these issues should be incorporated into the Conclusion statements

Reviewer 2 Report

In this paper, the authors evaluated radiologic, surgical and pathologic findings in order to determine the invasiveness of pituitary adenomas. 

  1. Since this is a general medicine journal, the Knosp classifications should be spelled out in more detail in Section 2.2. I think this is outlined in Table 1 but that table is a little confusing.  I would move that section out of this table and put it in separately in Section 2.2.
  2. The title of Table 2 should include a parenthesis that says (Classification #1). In the first columns of Tables 2 and 3, spell out Radiologic Invasiveness, Surgical Invasiveness, and Histologic invasiveness, cavernous sinus invasion and  sphenoid sinus invasion rather than the abbreviations and the footnotes – Just easier for the reader.

Reviewer 3 Report

Dear authors,

thank you for this interesting paper. Here my comments:

  • Please performa a language revision
  • it would be interesting to evaluate the vascular invasiveness. do you have any data on this?
  • about the discussion section, please provide subheadings, shorten the section, discuss about the diffenretial diagnosis of invasive sellar lesions 10.23736/S0390-5616.20.04931-0; discuss about the pathological findings including the histological vascular alterations https://pubmed.ncbi.nlm.nih.gov/2259404/ and the role of radiosurgery for those lesions 10.1007/s11102-022-01219-x

Congratulations

Author Response

This manuscript is a resubmission of an earlier submission. The following is a list of the peer review reports and author responses from that submission.